# Compression Property of TPEE-3D Fibrous Material and Its Application in Mattress Structural Layer

**DOI:** 10.3390/polym15183681

**Published:** 2023-09-07

**Authors:** Jiao-Jiao Fang, Li-Ming Shen

**Affiliations:** 1College of Furnishings and Industrial Design, Nanjing Forestry University, Nanjing 210037, China; jenna@njfu.edu.cn; 2Jiangsu Co-Innovation Center of Efficient Processing and Utilization of Forest Resources, Nanjing Forestry University, Nanjing 210037, China

**Keywords:** TPEE-3D fibrous material, compression property, mattress material, mattress structure layer, mattress total firmness, mattress multilevel firmness

## Abstract

Thermoplastic poly(ether/ester) elastomer (TPEE) has great potential as a mattress material due to its high resilience, breathability, and light weight. This study aimed to evaluate the feasibility of TPEE-3D fibrous material (T_3D_F), a three-dimensional block material made of TPEE fibers randomly aligned and loop-connected, for mattress application. After testing the compression properties of T_3D_F, the effects of T_3D_F structural layers on mattress firmness were investigated. The results showed that T_3D_F had good energy absorption capacity, broad indentation hardness range (126.94–333.82 N), and high compression deflection coefficient (2.79–4.39). The thickness and density of T_3D_F were the main factors influencing mattress firmness, and the impact of thickness was more significant (*p* < 0.05). Owing to the hard and soft segments contained in TPEE, T_3D_F could be used for both the padding and core layers of the mattress. The hardness value and $D_{surface}$ of the mattress with a T_3D_F padding layer increased with T_3D_F density but decreased with T_3D_F thickness. Moreover, the hardness value and $D_{surface}$ of the mattress with a T_3D_F core layer increased with T_3D_F density, while with T_3D_F thickness, its $D_{surface}$ increased and $D_{bottom}$ decreased. Therefore, the thick and low-density T_3D_F padding layer could improve the comfort of the mattress surface, a thin T_3D_F core layer could satisfy both the softer surface and the firmer bottom of the mattress.

## 1. Introduction

Thermoplastic poly(ether/ester) elastomer (TPEE) is a block copolymer with a microphase-separated structure consisting of alternately arranged poly(butylene terephthalate) (PBT) rigid blocks and poly(tetramethylene glycol) (PTMG) flexible blocks linked by ester linkages [1,2] (Figure 1). The rigid segment gives TPEE the same processing properties as thermoplastic; the soft segment gives the good elasticity of rubber. By adjusting the ratio of ester/ether segments, the hardness or other properties of TPEE can be changed to meet specific requirements. In addition, TPEE has excellent durability, heat resistance, and light weight [3]. Due to its multi-functional versatility, TPEE is widely used in automotive, electronic, transportation, and medical devices, such as shock-absorbing pads for high-speed railway tracks, hydraulic hoses, and artificial heart valves, making it an ideal material for multi-cycle load application conditions [4,5,6,7]. Among them, the three-dimensional TPEE fibrous material (abbreviated as T_3D_F) formed by the random arrangement of TPEE fibers in a loop connection has also brought innovation to the mattress industry (Beathair^®^ by Toyobo (Osaka, Japan), airfiber^®^ by Airweave (Tokyo, Japan); fullair^®^ by Keyi (Wuxi, China)) (Figure 2).

Mattresses are the direct support for human sleep and rest; the characteristics of the mattress have an enormous impact on sleep [8,9,10,11]. Mattress materials are the basic building blocks of a mattress. From the point of view of mattress structure, mattress materials are mainly divided into core layer materials and padding layer materials. Common core layer materials include spring, hard foam, palm, etc., while padding layer materials include polyurethane foam, natural latex, slow rebound foam, etc. [12,13]. Some researchers developed new constructions of hybrid springs for mattresses and demonstrated that they could achieve the desired progressive stiffness characteristics [14,15]. Chen studied the compression properties of luffa mattresses and found that compressing the LCs to the densification stage increased the firmness of the low-density luffa mattresses [16,17]. Compared to polyurethane mattresses, latex mattresses reduced peak body pressure and achieved a more even distribution of pressure [18], while significantly alleviating back pain and sclerosis in humans [19,20], attributed to the superior cushioning and resilience properties of latex. Moreover, variation in material parameters also affects mattress performance. Yu-Chi found that a 30 mm thickness of padding layer on the mattress was the most appropriate [21]. It is clear that the properties of a material depend largely on its response to loading, that better material properties could optimize mattress performance, and that mattress performance is significantly related to the type of materials, the choice of their parameters, and the way they are combined. 

T_3D_F has the potential to be a good mattress material because it not only retains the high elasticity and durability of TPEE, but also the large amount of pore space in its structure makes it highly breathable and easier to clean than ordinary polyurethane foam. Previous studies have shown that T_3D_F used in mattresses could improve pressure dispersion, facilitate turning movements and blood flow [22], and reduce the risk of pressure ulcers in long-term bedridden groups [23,24]. Moreover, these types of mattresses were found to be effective in lowering the body temperature of subjects during sleep, allowing the body to enter deep sleep earlier [25]. These studies aimed to evaluate the performance of this mattress at the human–mattress interface, often by comparing it to mattresses made from other materials to quantify sleep comfort after using T_3D_F. However, the results were far from conclusive, in part because they did not fully consider mattress performance, particularly the mattress firmness, which in turn affects the comfort of the human–bed interface. For this reason, further systematic research is needed to determine whether T_3D_F meets the design requirements for mattress firmness and whether it has more performance advantages than conventional structural layer materials.

Therefore, this study analyzed T_3D_F with different densities and thicknesses. Firstly, the compression properties of T_3D_F with different parameters were characterized. Secondly, the effect of T_3D_F on mattress firmness was investigated when used as a padding layer or a core layer. Meanwhile, a comparative analysis of mattress firmness was carried out by combining the commonly used pocketed spring, polyurethane foam, and latex. This study can provide a theoretical basis for the reasonable application of T_3D_F in mattresses.

## 2. Materials and Methods

### 2.1. Material and Combination of Mattress

The structure of the experimental mattresses was composed of three parts from top to bottom: a composite fabric layer, a padding layer, and a core layer. The composite fabric layer was identical for all mattresses and consisted of textile fabric, wadding fibers, and non-woven fabric quilting, with a thickness of 20 mm. The padding layer was a single layer of material selected from TPEE-3D fibrous material (T_3D_F), latex foam (LF), and polyurethane foam (PU). Two materials, T_3D_F and pocketed spring (PS), were used in the core layer respectively. In addition, T_3D_F was available in four densities and four thicknesses, PS was available in two spring parameters, while LF and PU were both available in one size. In total, 44 mattresses were combined. The sectional view of the mattresses is shown in Figure 3 and the detailed material parameters are listed in Table 1.

### 2.2. Measurement and Evaluation

#### 2.2.1. Testing of Material Properties

(1)Quasi-static compression

In order to determine the mechanical parameters of the different materials in the case of quasi-static behavior, a series of uniaxial compression experiments were carried out using an AG-X20KN universal mechanical testing machine (Shimadzu, Kyoto, Japan) (Figure 4). The loading pad was cylindrical in shape with a diameter of 100 mm and was pressed into the sample at a rate of 3 mm·min^−1^ to 80% of its thickness [26]. To ensure reliable and statistical results, the sample size was 80 mm × 80 mm × 80 mm. Five specimens were used for each type of material and each specimen was compressed only once. It has been found that environmental conditions have a major influence on the mechanical behaviors of materials [26,27]. In this study, all specimens used were tested in the same environment of 25 °C and 50% humidity, and each specimen was required to remain in this environment for 24 h prior to testing.

The Young’s modulus reflects the ability of a material to resist elastic deformation, the higher the value, the greater the stiffness of the material. The Young’s modulus (*E*) of a material is derived from a linear fit of the stress–strain data for each specimen using the linear regression method [28]. 

The energy absorption capacity of the material is an important aspect that affects its cushioning performance when used in mattresses, which is mainly assessed by the two indices of absorbed energy and energy absorption efficiency. During compression, the area between the stress–strain curve and the strain axis of the material is the absorbed energy by the material, expressed in Equation (1) [29]. The energy absorption efficiency (*E_ea_*) is the ratio of the absorbed energy by the material to the corresponding stress, given by Equation (2) [30,31].
(1)$$W=\int_{0}^{\varepsilon_{a}}\sigma \left(\varepsilon \right)d\varepsilon ,\;0\leq \sigma_{a}\leq 0.8$$
(2)$$E_{ea}=\frac{\int_{0}^{\varepsilon_{a}}\sigma \left(\varepsilon \right)d\varepsilon }{\sigma_{a}},\;0\leq \sigma_{a}\leq 0.8$$
where *W* is the absorbed energy (J); $E_{ea}$ is the energy absorption efficiency (%); $\varepsilon_{a}$ is the compaction strain; ε is the arbitrary strain; *σ*(ε) is the stress corresponding to the strain (MPa); $\sigma_{a}$ is the stress corresponding to the compaction strain (MPa). 

(2)Indentation property of the material

The indentation property of the material was measured according to ISO 2439:2008(E) [32]. The test was performed using the same machine (AG-X20KN) with a cylindrical indenter of 200 mm diameter. The loading speed was 100 mm·min^−1^ and the maximum compression was 75% of the initial thickness. The sample size was 400 mm × 400 mm × 50 mm, with 5 duplicate samples for each material. The indentation hardness and the compressive deflection coefficient (*S_f_*) of the material were determined from the load-deflection curve obtained from the test.

Indentation hardness: the corresponding force after maintaining for the 30 s when the sample was indented to 40% of its thickness.Compressive deflection coefficient: *S_f_* = *F*65/*F*25, is the ratio of the force at 65% and 25% indentation in compression.

#### 2.2.2. Testing of Mattress Firmness

(1)Total firmness of the mattress

The total firmness of the mattress was tested according to the ISO 23769: 2021 standard [33]. The testing machine was the same as described above. The loading pad was a ball-capped rigid object with a diameter of 355 mm and a contact area of 1000 cm^2^ (Figure 5). The loading speed was 90 mm·min^−1^ and the force range was 0–1000 N. The mattress sample had a width dimension of 500 mm × 500 mm and 5 replicates per mattress. Prior to the test, the mattress had to be kept at room temperature of 25 °C and 50% humidity for 24 h.

The hardness value (H) and the firmness rating (H_s_) are two indicators of the total firmness of a mattress, both determined from measured load–deflection curves. 

Hardness value: the average of the slopes of the load–deflection curves at 210 N, 275 N, and 340 N, as shown in Equation (3) [33], the unit is N/mm.
(3)$$H=\frac{C_{210}+C_{275}+C_{340}}{3}$$Firmness rating: a number (1 decimal) from 1 to 10 characterizes the degree of firmness of the mattress, from firm to soft, as calculated in Equations (4) and (5) [33].

(4)$$H_{s}=10{\left\{1-{exp}^{-\left(\frac{K}{a}+b\right)}\right\}}^{2}$$(5)$$K=\frac{A_{450}}{H}$$
where *a* = 1/5.92 × 10^−4^ mm^2^, *b* = 0.148, *A*_450_ is the area under the load–deflection curve (0–450 N).

(2)Multilevel firmness of the mattress

After 24 h in place, the mattress samples were tested on a universal mechanical machine (AG-X20KN) using a 100 mm diameter cylindrical indenter. The load range was 0–250 N at a constant speed of 100 mm min^−1^. Based on the load-deflection curves obtained from the tests, the approximate modulus calculation method was used to determine the multilevel firmness of the mattress [17]: surface firmness ($D_{surface}$), core firmness ($D_{core}$), and bottom firmness ($D_{bottom}$), as shown in Equations (6)–(8), in MPa.
(6)$$D_{surface}=\frac{36/S}{\varepsilon_{f40N}-\varepsilon_{f4N}}$$
(7)$$D_{core}=\frac{160/S}{\varepsilon_{f200N}-\varepsilon_{f40N}}$$
(8)$$D_{bottom}=\frac{50/S}{\varepsilon_{f250N}-\varepsilon_{f200N}}$$
where $D_{surface}$, $D_{core}$, and $D_{bottom}$ represent the modulus between 4 and 40 N (surface), 40 and 200 N (core), 200 N and 250 N (bottom); ε represents the strain at 4 N, 40 N, 200 N, and 250 N; *S* is the bottom area of the indenter.

### 2.3. Statistical Analysis

For the data obtained from the tests, the normal distribution was first analyzed for significance using the Kolmogorov–Smirnov test (sig. > 0.05). One-way analysis of variance (ANOVA) was used to analyze the differences in each index between T_3D_F of different densities and between other materials. In addition, a two-way ANOVA was used to determine the main effects of density and thickness on mattress firmness and their interactions when T_3D_F was used for the mattress structural layer. Duncan’s test was performed separately to determine significant differences in mattress firmness within the density and thickness groups. Mattress firmness was also compared between the use of T_3D_F and the use of other regular materials. Statistical analyses were performed using SPSS 20.0 statistics software (Chicago, IL, USA) at the 5% level of significance.

## 3. Results

### 3.1. Compression Properties of the Materials

#### 3.1.1. TPEE-3D Fibrous Material

The non-linear curves showed the viscoelasticity of T_3D_F (Figure 6a). The compression curves gradually transitioned from the elastic deformation phase (phase I) to the plastic strengthening phase (phase II) and compacting phase (phase III), which deviated from Hooke’s law gradually. The mesh skeleton of T_3D_F, the main stressed part, first underwent elastic deformation. Due to the small strain range in this phase (0–0.13), T_3D_F would return to its original state after stress relief. The compressive *E* of T_3D_F ranged from 0.016 MPa to 0.044 MPa, which increased with increasing density (Figure 6b), and the difference in *E* between any two densities was statistically significant (*p* < 0.001).

In phase II, the T_3D_F mesh skeleton underwent partial plastic deformation due to the high stress, and its polyester fibers began to be stressed simultaneously. During the deformation process, some of the stress was consumed, resulting in a rapid increase in strain and a relatively small increase in stress. When the compressive stress on T_3D_F reached a certain value, the mesh skeleton was completely deformed, the stress increased sharply with the strain, the pores within the skeleton were compacted, and the curve entered phase III. Moreover, the strain range of phase II decreased as the density of T_3D_F increased, meaning that the denser T_3D_F entered phase III earlier. At the same strain, the compressive stress gradually increased with increasing T_3D_F density, indicating that the denser T_3D_F was stiffer, i.e., the polyester fibers forming the skeleton were stiffer and could withstand greater stress.

Figure 7a shows that the absorbed energy of T_3D_F decreased with increasing density in the stress range of 0–0.005 MPa and gradually increased when the stress exceeded 0.009 MPa. However, there was no regularity in the variation of absorbed energy with density among the stress range of 0.005–0.009 MPa. Due to the large strain range (0.13–0.63), phase II was the main phase of energy absorption and the increments of T_3D_F stress and absorbed energy were calculated in Figure 7b. At the same strain of T_3D_F, the higher the stress, the more energy was absorbed, so 75 kg m^−3^ T_3D_F (referred to as T_3D_F 75) absorbed the most energy in phase II with an increase of 0.0038 J.

The *E_ea_* values of T_3D_F tended to increase and then decrease with the stress (Figure 7c), suggesting the existence of a peak value at which the energy absorption capacity of T_3D_F was optimal. Figure 7d shows that the maximum *E_ea_* value of T_3D_F increased with density, and the stresses corresponding to the maximum *E_ea_* values for four densities of T_3D_F (45–75 kg·m^−3^) were 0.004 MPa, 0.007 MPa, 0.008 MPa, and 0.011 MPa, respectively. 

Table 2 lists the results of the indentation properties of T_3D_F. The indentation hardness ranged from 126.94 N to 333.82 N and increased with density. T_3D_F 45 was the softest, in agreement with the quasi-static compression results described above. The *S_f_* value characterized the deformation resistance of the material, and a higher *S_f_* value indicated a softer surface or harder base. The *S_f_* value of T_3D_F decreased with increasing density and all *S_f_* values were greater than 2.5. Therefore, the surface of low-density T_3D_F was softer and the base of high-density T_3D_F was harder. 

#### 3.1.2. Comparison of the Properties between T_3D_F and Other Materials

The stress–strain curves of PU, LF, and T_3D_F 45 (Figure 8a) all correspond to the three phases of compression of porous materials: phase I, elastic deformation; phase II, plastic strengthening; and phase III, compaction. However, the change in each material during compression was not uniform and at the same strain, the materials could be in different phases of deformation. The three phases of the PU curve were the easiest to distinguish, particularly phase II, where the curve converged to a horizontal line, with a smaller increase in stress but a greater deformation in compression, in contrast to the other two materials. For the curves of LF and T_3D_F 45, both phases I were present only in small strain segments, while the stresses in phase II were relatively more variable. Specifically, when the strain was less than 0.28, the PU could withstand more stress at the same strain. In the strain range of 0.28–0.36, T_3D_F 45 maintained a high stress level. Then, the stress of LF remained higher after strain exceeded 0.36. In addition, the *E* of PU was significantly higher than that of T_3D_F 45 and LF (*p* < 0.001), whereas there was no significant difference between the *E* of T_3D_F 45 and LF (*p* = 0.065) (Figure 8b). 

Figure 9a indicates that the energy absorbed by the materials varied with the stress. LF absorbed the most energy when the stress was less than 0.0023 MPa, whereas in the range 0.0023–0.0072 MPa, the comparison of energy absorbed was PU > T_3D_F 45 > LF. The pattern of *E_ea_* was consistent with the energy absorbed (Figure 9b). PU had the highest *E_ea_* (0.44%) at the stress of 0.0033 MPa, followed by T_3D_F 45 with a peak *E_ea_* of 0.32%. The results implied that the maximum *E_ea_* of PU was the highest, but it occurred at lower stresses. In contrast, T_3D_F could absorb more energy at higher stress and more energy could be absorbed at higher density (Figure 7a), implying that the energy absorption capacity of T_3D_F was significantly better than that of PU and LF.

As shown in Table 3, the indentation hardness of PU was the highest, followed by LF, while that of T_3D_F 45 was the lowest (126.94 N). In addition, LF had the highest *S_f_* value and 25% lower compression force (69.33 N), meaning that LF had the softest surface. ANOVA showed that the difference in *S_f_* values between T_3D_F 45 and LF was not significant (*p* = 0.082), but both were significantly greater than PU (*p* < 0.05), indicating that the *S_f_* value of T_3D_F 45 was comparable to that of LF and superior to that of PU.

### 3.2. TPEE-3D Fibrous Material for Mattress Padding Layers

#### 3.2.1. Effect of the T_3D_F Padding Layer on Mattress Firmness

Table 4 shows that the density and thickness of the T_3D_F padding layer had a significant effect on mattress firmness (*p* < 0.05), but there was no interaction between them (*p* = 0.148), i.e., the influence of thickness on mattress firmness did not differ between densities. From the *F*-value, the influence of thickness on mattress firmness was greater than that of density, indicating the importance of thickness selection in the structure of the mattress padding layer. As shown by the between-subjects effect test (Table 5), both density and thickness of T_3D_F had a significant effect (*p* < 0.05) on the hardness value, firmness rating, $D_{surface}$, and $D_{bottom}$ of the mattress, while the greatest influence was found on $D_{surface}$ (*F*_D_ = 9.834, *F*_T_ = 14.849). Therefore, the effects of the thickness and density of the T_3D_F padding layers on the mattress firmness were compared according to the mean of their main effects.

Under the T_3D_F padding layer of different densities, the mattress firmness is shown in Figure 10. As the density increased, the hardness value increased and the firmness rating decreased (Figure 10a), the mattress became firmer. The difference between the low-density (45–55 kg·m^−3^) and high-density (65–75 kg·m^−3^) T_3D_F padding layer was significant (*p* < 0.05) for both indices. In addition, the firmness rating of mattresses ranged from 2.3 to 5.3, indicating that with the same core layer (PS: 2.0 60 6/80), the mattress could be configured with different firmness, from moderately soft to firm, with the available density of T_3D_F padding layers. 

The multilevel firmness of the mattress gradually increased from the surface to the bottom (Figure 10b). As the density of the T_3D_F padding layer increased, the $D_{surface}$ of the mattress increased, while the $D_{core}$ and $D_{bottom}$ decreased. The mattress with the T_3D_F 45 padding layer had the softest $D_{surface}$ and the firmest $D_{core}$ and $D_{bottom}$, which could be influenced by the compression process of the mattress. The softer the surface (T_3D_F 45), the greater the deformation of the mattress; after continued compression, the indenter came into contact with the spring core layer zone in advance, making the compressible thickness of the mattress smaller; and the spring’s resistance to deformation increased with the force, making the mattress require more force to deform. Duncan’s test revealed that the $D_{surface}$ of the mattress with T_3D_F 45 padding layer was significantly lower than that of T_3D_F 65 and T_3D_F 75 (*p* = 0.020, *p* = 0.000), and the difference between the $D_{surface}$ of T_3D_F 55 and T_3D_F 75 was also significant (*p* = 0.005). However, there was no significant difference between $D_{bottom}$ of T_3D_F 55 and T_3D_F 65 (*p* = 0.919), while the remaining two densities were significantly different (*p* < 0.05). 

Figure 11 shows the mattress firmness between two thicknesses of the T_3D_F padding layer. As the thickness increased from 20 mm to 40 mm, the hardness value decreased and the firmness rating increased (Figure 11a), indicating that the total firmness decreased with the padding thickness and there was a significant difference in hardness values between two thicknesses (*p* = 0.009). Moreover, $D_{surface}$ and $D_{bottom}$ decreased significantly with thickness (*p* = 0.006, *p* = 0.039), while $D_{core}$ was similar (*p* = 0.176) (Figure 10b). In general, increasing the thickness of the padding layer enhanced the softness of the mattress.

#### 3.2.2. Comparison of Mattress Firmness between T_3D_F and Other Padding Materials

The differences in mattress firmness between single material PU, LF, and T_3D_F padding layers are illustrated in Figure 12 when the core layers were all PS (2.0 60 6). The hardness value was T_3D_F 75 > PU > LF > T_3D_F 45, with the firmness rating reversed (Figure 12a). This indicated that the mattress with the T_3D_F 45 padding layer had the lowest total firmness and belonged to the medium-firm mattresses (H_s_ = 5.8). The ANOVA showed that the differences in total firmness between the different padding materials were significant (*p* < 0.05), except for LF and PU (*p* > 0.05).

Figure 12b shows that the $D_{surface}$ of the mattress between different padding materials was PU > T_3D_F 75 > LF > T_3D_F 45; the $D_{surface}$ of the mattress with PU padding layer was 2.02 times higher than that of T_3D_F 45 and was significantly different from other padding layers (*p* < 0.05). In addition, the $D_{surface}$ of the mattress was comparable between T_3D_F 75 and LF padding layer, and the $D_{surface}$ of the mattress with T_3D_F 45 padding layer was significantly lower than that of LF (*p* = 0.009). Therefore, the use of the T_3D_F padding layer could significantly reduce the $D_{surface}$ of the spring mattress, and the mattress surface was softer than LF when low-density T_3D_F was used. 

The $D_{core}$ of the mattresses with different padding materials was T_3D_F 45 > LF > T_3D_F 75 > PU, and the $D_{core}$ of the mattress with T_3D_F 45 padding layer was significantly higher than that of the PU (*p* = 0.017), which was influenced by the $D_{surface}$. The softer surface resulted in more deformation, so that more force was required to deform the mattress further, whereas the deformation of the mattress was relatively less for the same force, i.e., the mattress was firmer. The change in $D_{bottom}$ was similar to that of $D_{core}$. On the other hand, T_3D_F 45 was an excellent material for the padding layer, even superior to LF. 

### 3.3. TPEE-3D Fibrous Material for Mattress Core Layers

#### 3.3.1. Effect of the T_3D_F Core Layer on Mattress Firmness

When T_3D_F was used for the mattress core layer, the density, thickness, and their interaction all had a significant effect on mattress firmness (*p* < 0.05) (Table 6), and the influence of thickness was greater than that of density (*F* = 208.981). This meant that thickness was an important factor and the effect of thickness on mattress firmness varied across densities. Further checking the results of the main effects (Table 7), thickness and density had a significant effect on all indicators of mattress firmness (*p* < 0.05), while the interaction only had a significant effect on $D_{core}$ and $D_{bottom}$ (*p* < 0.05). In addition, the *F*-value shows that thickness and density had the greatest effect on $D_{bottom}$, while the interaction had the greatest effect on $D_{core}$. Therefore, the effects of density and thickness on hardness value, firmness rating, and $D_{surface}$ were directly based on mean comparisons of main effects, while their effects on $D_{core}$ and $D_{bottom}$ were analyzed by means of their individual effects.

Figure 13 shows the effect of the density and thickness of T_3D_F core layer on the hardness value, firmness rating, and $D_{surface}$. As the density increased, the hardness value and $D_{surface}$ increased while the firmness rating decreased, i.e., the mattress’s total firmness became firmer (Figure 13a). The differences in hardness value or $D_{surface}$ between four densities were not significant (*p* = 0.062, *p* = 0.499). However, the mattress with T_3D_F 45 core layer had a significantly higher firmness rating than the others (*p* < 0.05). Furthermore, as the thickness of the T_3D_F core layer increased, the hardness value decreased while the firmness rating and $D_{surface}$ increased (Figure 13b). All three indices were significantly different between different thicknesses (*p* < 0.05). This indicates that the thicker the T_3D_F core layer, the softer the total firmness and the firmer the mattress surface. Moreover, the mattresses with a T_3D_F core layer were all classified as firm mattresses (H_s_: 0–2).

When the T_3D_F density ranged from 45 kg·m^−3^ to 75 kg·m^−3^, the differences in $D_{core}$ and $D_{bottom}$ of mattresses with three thicknesses of T_3D_F core layers are shown in Figure 14. In the case of T_3D_F 45 core layer, $D_{core}$ and $D_{bottom}$ decreased with increasing thickness, indicating that the mattress with the thicker core layer had a softer core and bottom, while for other densities, only $D_{bottom}$ was negatively correlated with thickness. The comparisons showed that all $D_{core}$ and $D_{bottom}$ of the mattresses were significantly different between the core layer thicknesses (*p* < 0.05), except for the $D_{core}$ of the T_3D_F 55 core layer (*p* > 0.05).

Figure 15 shows the variation of $D_{core}$ and $D_{bottom}$ of the mattress among the four densities of T_3D_F core layers at each thickness. When the core layer thickness was 80 mm, the $D_{core}$ of the mattress gradually increased with the density, and there were significant differences among the four densities (*p* < 0.05) (Figure 15a). However, when the thickness was 40 or 60 mm, the effect of the density of the core layer on $D_{core}$ of the mattress was not remarkable (*p* > 0.05). This could be influenced by the structure of T_3D_F, which consisted of strips of hollow filaments arranged in a random longitudinal and horizontal pattern; when T_3D_F was cut into different thicknesses, the size and distribution of the pores on different cross-sections were different (Figure 2), which led to the uneven density distribution of T_3D_F, and the smaller the thickness, the greater the porosity, which in turn affected the hardness of the material and its mattress firmness. Therefore, the influence of the core layer density on the mattress decreases as the thickness decreases.

The $D_{bottom}$ (Figure 15b) of the mattress decreased with increasing density at thicknesses of 40 mm and 60 mm, but it does not have a linear law with the change in density at thickness of 80 mm, which could be influenced by the $D_{bottom}$. ANOVA showed that the effect of density on $D_{bottom}$ was statistically significant at all thicknesses (*p* < 0.05). Moreover, the mattress always had the highest $D_{bottom}$ in the T_3D_F 45 core layer, which was consistent with the results for the T_3D_F padding layer.

#### 3.3.2. Comparison of Mattress Firmness between T_3D_F and Other Core Materials

The mattress firmness of the different core layers (PS versus T_3D_F) was compared at the same padding layers (LF, 40 mm) (Figure 16). The comparison of the hardness value showed T_3D_F 75 > PS 2.18 > T_3D_F 45 > PS 2.08, with the firmness rating reversed. The total firmness of the mattress differed significantly (*p* < 0.05) between the core layers. 

Regarding the multilevel firmness of the mattress, the $D_{surface}$ of the four core layers was similar, while the $D_{core}$ and $D_{bottom}$ were significantly different (*p* < 0.05). The $D_{core}$ of the T_3D_F 75 core layer mattress was significantly higher (*p* < 0.05), which may be influenced by the hardness of the core layer material. Due to the change in strain of the mattress during the compression process, the $D_{bottom}$ was the lowest. 

## 4. Discussion

The mattress structure, the material of the mattress and its combination, directly affects the firmness of the mattress, which is an important aspect affecting the comfort of the human–mattress interface [34]. In this study, T_3D_F was introduced as the structural layer material of the mattress. Taking advantage of the good energy absorption capacity and larger compression deflection coefficient (*S_f_*) of T_3D_F, it performed well in both the mattress padding and core layers. Experiments suggested that a reasonable choice of T_3D_F thickness and the structural layer used for it could better regulate the mattress firmness. Thus, T_3D_F was a preferable substitute for mattress material. With the progress of science and the continuous improvement of material properties, the application of T_3D_F in mattresses will be continuously expanded.

Energy absorption capacity is an important parameter that visually reflects the cushioning performance of the material. Mattress materials with better cushioning performance can absorb more impact energy and keep the stress transmitted to the body at a lower threshold [35,36]. This study revealed that the peak energy absorption of T_3D_F ranged from 0.0028 to 0.0085 J and increased with increasing density (Figure 7a). The best energy absorption of T_3D_F was found when the stress range was 0.004–0.011 MPa (Figure 7c). The difference in energy absorption of the material was related to its deformation mechanism. When T_3D_F was compressed, as the density increased, the greater the strength of the individual fibers that formed the T_3D_F mesh skeleton, the greater the stress it could withstand and the higher the energy absorption capacity. Through the energy absorption diagram, T_3D_F with the optimum energy absorption capacity could be selected for different stress limits. 

In addition, the energy absorption capacity of T_3D_F was significantly higher than PU and better than LF, except for T_3D_F 45 (Figure 9), which was mainly attributed to the large amount of pore space in the T_3D_F structure [37]. From loofah, which also contained many pores, it was found that the low-density loofah could absorb higher energy in the range of 0.01–0.035 MPa with a maximum *E_ea_* of about 0.4 [17]. However, the loofah treated with compression absorbed more energy due to its higher stress (0–0.56 MPa). In contrast, the stress range of loofah was almost seven times higher than that of T_3D_F, but the maximum *E_ea_* increased by only 25% (0.32 versus 0.4), indicating that the energy absorption capacity of T_3D_F remained superior for the same stress range. On the other hand, T_3D_F can also be recovered and degraded for recycling, reducing plastic emissions, and ensuring environmental sustainability.

A higher *S_f_* value indicated a softer surface or a harder bottom of the material, which was directly related to the support capacity at different load levels [38,39]. The *S_f_* values of T_3D_F ranged from 2.9 to 4.3 and decreased with increasing density (Table 2). This meant that when T_3D_F was used for the mattress, its performance in reducing body pressure gradually decreased with density, but the resistance of the mattress to the bottom gradually increased after loading. Therefore, T_3D_F could be used not only as a padding layer to meet the softness requirement in contact with the human body, but also as a core layer with sufficient hardness to provide good body support. 

Compared to PU and LF, the *S_f_* value of T_3D_F 45 was comparable to LF and significantly higher than PU (Table 3). Scarfato found the *S_f_* value of PU to be 2.7 [39], similar to the present study. Thus, the low-density of T_3D_F was more suitable for use as a padding layer, the same as the LF, providing a softer surface for mattresses than that of PU.

The mattress padding layer was in close contact with the human body and soft padding materials were usually used to improve comfort, reduce pressure on bony prominences, and prevent pressure sores [40]. When T_3D_F was used as the padding layer, its density and thickness were important factors influencing mattress firmness (Table 5), with a greater effect of thickness, especially on the $D_{surface}$. A moderate increase in the thickness of the padding layer would result in a softer mattress surface [21], which was consistent with the results of this study. In addition, the strain range of the tested surface firmness was 0–0.25, which was still within the range of the padding thickness; thus, the padding layer was the main load-bearing area. As the density increased, the indentation hardness of T_3D_F (Table 2) and the $D_{surface}$ of the mattress both increased (Figure 10b), implying that there was a correlation between the $D_{surface}$ of the mattress and the indentation hardness of the material. 

Comparing different materials, the mattress containing T_3D_F 75 padding layer had the highest $D_{surface}$, owing to the highest indentation hardness of T_3D_F 75. The $D_{surface}$ of the mattress with T_3D_F padding layer covered the range between PU and LF, and the $D_{surface}$ of T_3D_F 45 padding layer was softer than that of LF. Ren showed that the use of softer padding materials could effectively conform to the body and facilitate better shoulder and hip coverage [41]. Others found that the addition of a latex padding layer could effectively reduce mattress firmness compared to a firmer palm mattress [28]. Therefore, thick and low-density T_3D_F was more suitable as a padding layer to enhance the comfort of the mattress surface. Furthermore, the mattress with T_3D_F padding layer offered a wide range of firmness adjustments to meet the needs of different weight groups.

The core layer of the mattress was an important structural layer to support the human body, which not only required high firmness, but also needed good elasticity to mitigate the vibrations caused by the bed frame [42]. This study confirmed that the thickness of the T_3D_F core layer was the primary factor influencing the $D_{bottom}$ of the mattress, followed by T_3D_F density, and their interaction notably affected $D_{core}$ (Table 6 and Table 7). Increasing thickness of the core layer led to a gradual decrease in $D_{bottom}$. When a core layer with a thickness of 40 mm was used, the mattress had the lowest $D_{surface}$ and the highest $D_{bottom}$ (Figure 13a and Figure 15). Thus, the thickness of the core layer of the mattress does not appear to relate to better properties. A 40 mm T_3D_F core layer was sufficient for the mattress with a soft surface and a firm bottom.

When the thickness was 80 mm, the $D_{core}$ of the mattress increased with the density of the T_3D_F core layer (Figure 15a). Upon comparison with the pocketed spring, the mattress firmness was similar to that of PS 2.08 at T_3D_F 45 core layer, while T_3D_F 75 was greater than PS 2.18 (Figure 16). The results indicate that T_3D_F, when used as a mattress core layer, was comparable to the hardness range of the spring mattress and could meet the needs of the mattress firmness design. In addition, the mattress with a T_3D_F core layer displayed superior deformation performance compared to spring mattresses. This makes it suitable for a variety of bed sets or frames, particularly multifunctional ones that require high levels of mattress softness. It also aided in offering reasonable support to the human body in various sitting, lying, or semi-lying positions. 

Based on the analysis of the compression properties of T_3D_F and their impact on mattress firmness, a design strategy was proposed to meet the demand for regulating mattress firmness using T_3D_F. Firstly, the thickness of a mattress was the main factor affecting its firmness. The choice of thickness was influenced by the mattress structure layer. If T_3D_F is used as a padding layer, a moderate increase in thickness can lead to a softer surface and more even pressure dispersion. In the forthcoming studies on human–mattress interaction, the impact of T_3D_F thickness on interface pressure will be probed to identify the optimal configuration for padding thickness and mattress firmness. Furthermore, a thin T_3D_F core layer satisfied both the softest surface and the firmest bottom of the mattress. This design also increased the economic and practical benefits of the T_3D_F mattress. Moreover, to reduce the firmness rating of the mattress with a T_3D_F core layer, a softer padding material such as a slow rebound gel could be utilized in subsequent studies.

Secondly, the T_3D_F density was also an important parameter in mattress design that cannot be ignored. When the density ranges from 45 kg·m^−3^ to 75 kg·m^−3^, the E, indentation hardness, and *S_f_* values displayed significant differences across the four densities. If T_3D_F is used for the mattress padding layer, low-density T_3D_F with lower indentation hardness, higher energy absorption capacity, and higher *S_f_* value can be considered to make a mattress surface with better softness and cushioning. If T_3D_F is used as a core layer of the mattress, it will result in a firm mattress regardless of its density. Additionally, the mattress will significantly improve in terms of support and cushioning. In addition, the pressure at the human–mattress interface depended not only on the mattress firmness, but also on the weight of the individual [42,43]. Therefore, incorporating the weight of the participant in future studies would aid in the selection of a suitable density for the T_3D_F core layer. 

## 5. Conclusions

To fully comprehend the properties of T_3D_F and its impact on mattress firmness, four densities and four thicknesses of T_3D_F were incorporated for analysis. The mattress firmness was measured when T_3D_F was used as the padding layer and the core layer separately. The conclusions obtained are as follows: (1)T_3D_F had good energy absorption capacity, broad indentation hardness range, and higher *S_f_* value, making it an appropriate material for a mattress that can serve both as padding and core layers. (2)As a padding layer, T_3D_F’s thickness and density had a significant impact on $D_{surface}$ of the mattress. Using a thick and low-density T_3D_F proved to be a fitting option for enhancing the comfort of the mattress surface.(3)As a core layer, T_3D_F’s thickness and density had a notable impact on $D_{bottom}$ of the mattress, while their interaction effected the $D_{core}$ significantly. A thin T_3D_F core layer could achieve a softer surface and a firmer bottom of the mattress. (4)T_3D_F thickness was the main factor affecting mattress firmness, and the impact of thickness was related to its structural layer. For different structural layers, T_3D_F thickness should be selected reasonably based on the firmness requirements of the mattress.

## Figures and Tables

**Figure 1 polymers-15-03681-f001:**
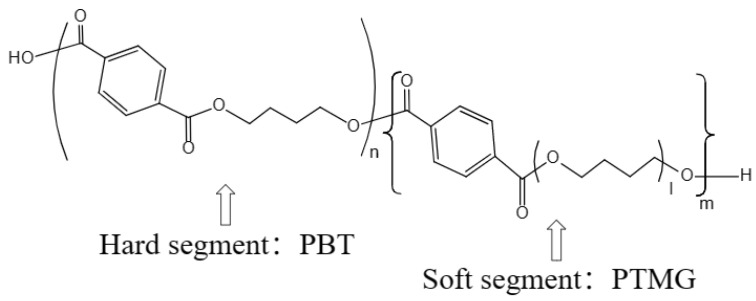
The chemical structure of thermoplastic poly(ether/ester) elastomer (TPEE).

**Figure 2 polymers-15-03681-f002:**
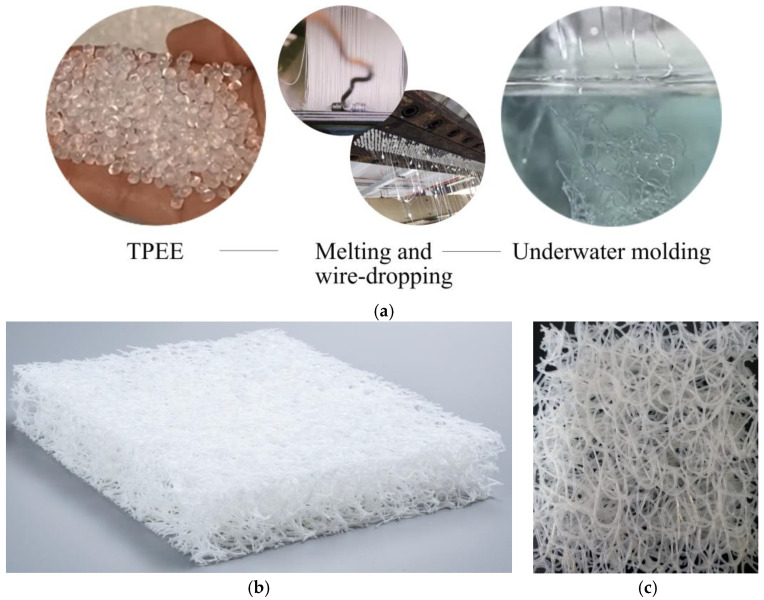
Illustration of TPEE-3D fibrous material (T_3D_F). (**a**) Molding process; (**b**) Perspective view, (**c**) Surface close-up.

**Figure 3 polymers-15-03681-f003:**
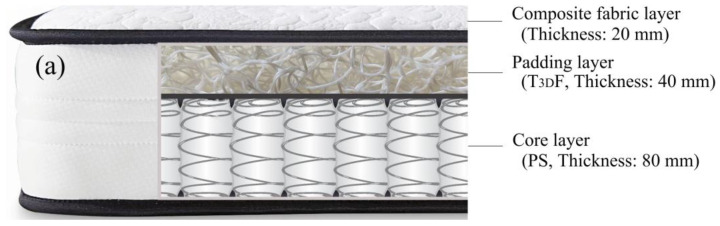
Construction of the mattresses. (**a**) TPEE-3D fibrous material (T_3D_F) used as a padding layer, (**b**) T_3D_F used as a core layer.

**Figure 4 polymers-15-03681-f004:**
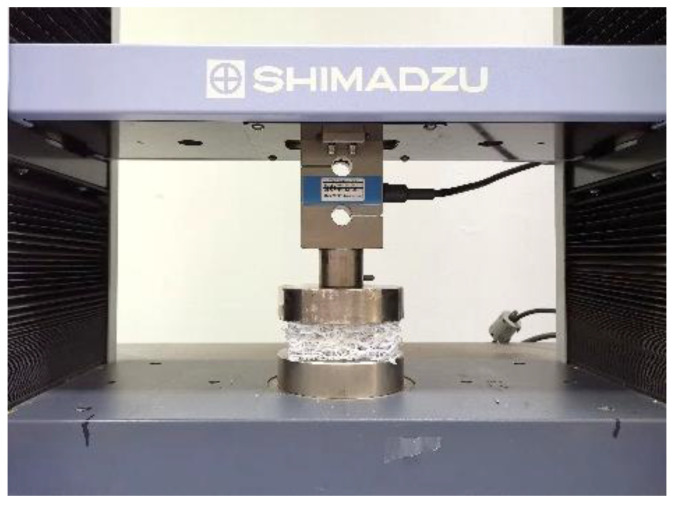
Setup for quasi-static compression.

**Figure 5 polymers-15-03681-f005:**
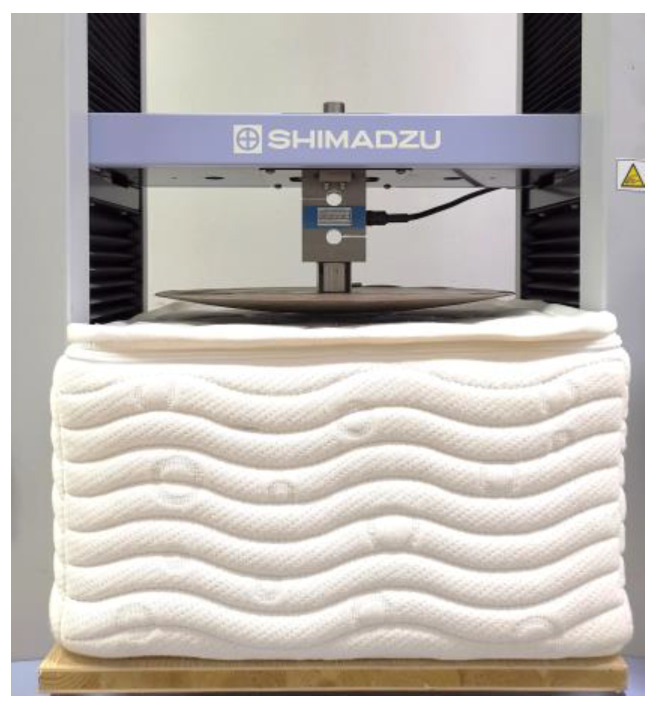
Setup for testing of mattress’s Total firmness.

**Figure 6 polymers-15-03681-f006:**
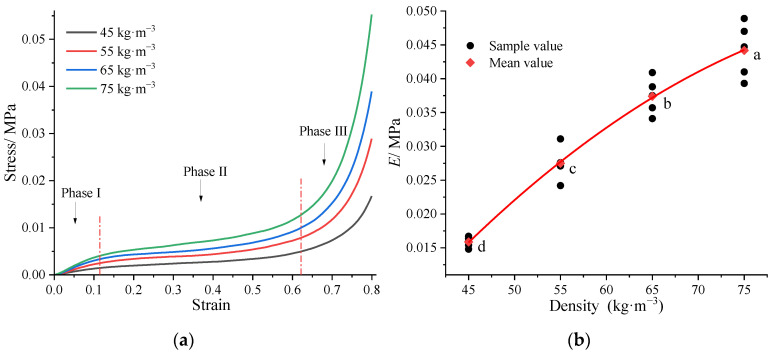
Results of quasi-static compression of T_3D_F. (**a**) Stress–strain curve. The stress–strain curve of T_3D_F during compression consists of three distinct phases: the elastic deformation phase (phase I), the plastic strengthening phase (phase II), and the compacting phase (phase III). (**b**) Young’s modulus (*E*). Variation of *E* of T_3D_F with its four densities. Significant differences between different letters in the graph are the result of multiple comparisons (*p* < 0.05).

**Figure 7 polymers-15-03681-f007:**
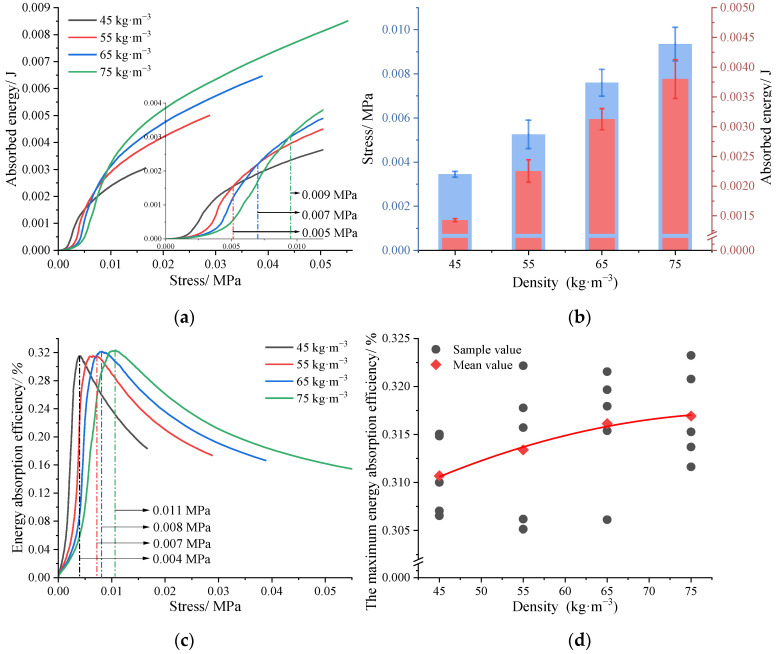
Energy absorption capacity of T_3D_F. (**a**) Absorbed energy; (**b**) Increment of stress and absorbed energy in phase II; (**c**) Energy absorption efficiency (*E_ea_*); (**d**) Variation of maximum *E_ea_* value with T_3D_F density.

**Figure 8 polymers-15-03681-f008:**
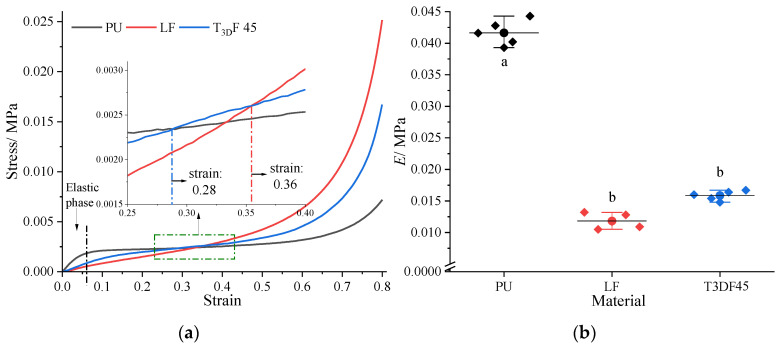
Test results of quasi-static compression of the material. (**a**) Stress–strain curve; (**b**) Young’s modulus (*E*). Significant differences between different letters in the graph are the result of multiple comparisons (*p* < 0.05).

**Figure 9 polymers-15-03681-f009:**
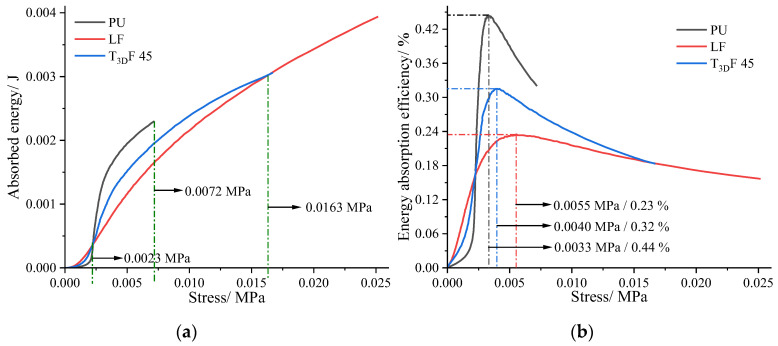
Energy absorption capacity of different materials. (**a**) Absorbed energy, (**b**) Energy absorption efficiency (*E_ea_*).

**Figure 10 polymers-15-03681-f010:**
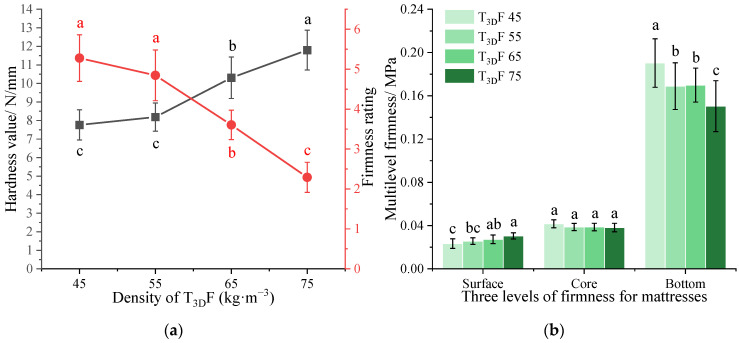
Mattress firmness of the T_3D_F padding layer at different densities. (**a**) Total firmness, (**b**) Multilevel firmness. Significant differences between different letters in the graph are the result of multiple comparisons (*p* < 0.05).

**Figure 11 polymers-15-03681-f011:**
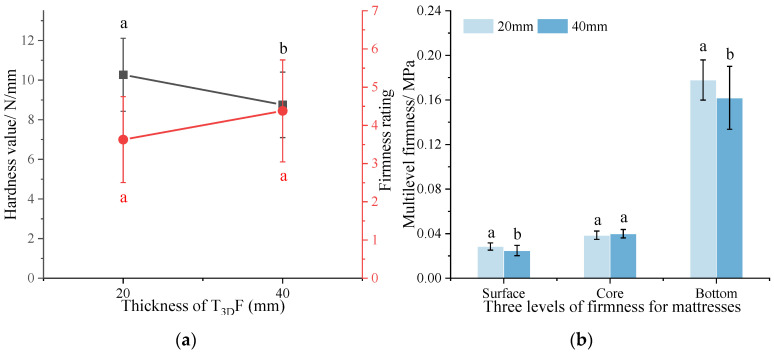
Mattress firmness of the T_3D_F padding layer at two different thicknesses. (**a**) Total firmness, (**b**) Multilevel firmness. Significant differences between different letters in the graph are the result of multiple comparisons (*p* < 0.05).

**Figure 12 polymers-15-03681-f012:**
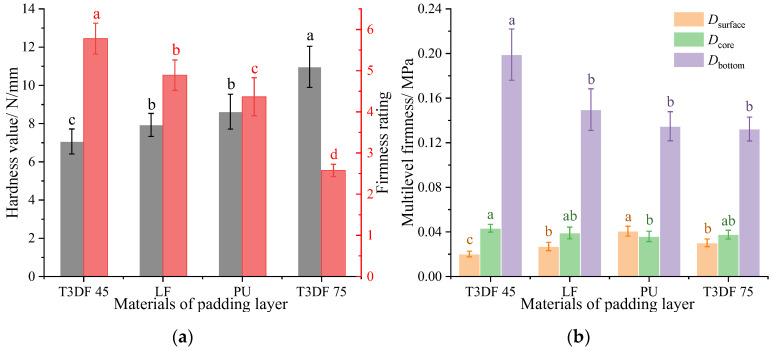
Mattress firmness of different padding layers. (**a**) Total firmness, (**b**) Multilevel firmness. Significant differences between different letters in the graph are the result of multiple comparisons (*p* < 0.05).

**Figure 13 polymers-15-03681-f013:**
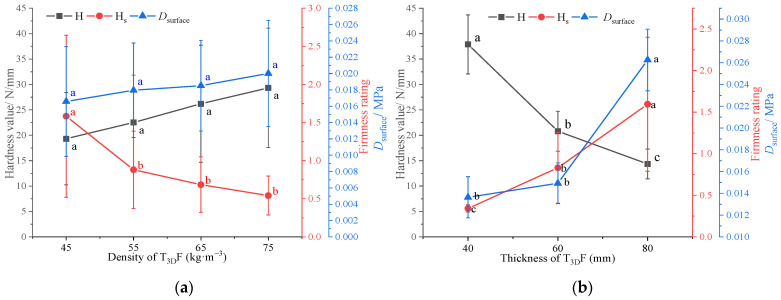
Effect of core density (**a**) and thickness (**b**) on hardness value, firmness rating, and $D_{surface}$ of the mattress. Significant differences between different letters in the graph are the result of multiple comparisons (*p* < 0.05).

**Figure 14 polymers-15-03681-f014:**
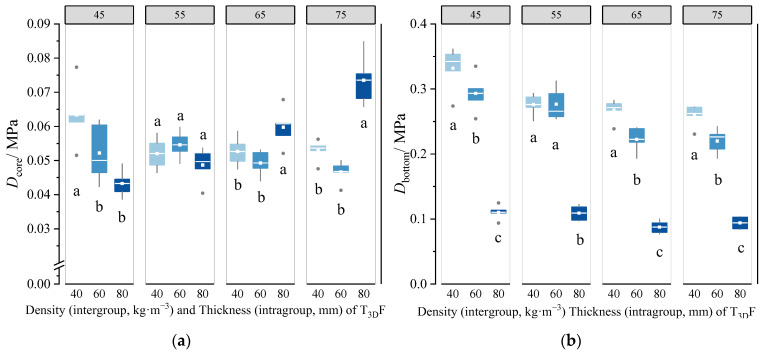
Effect of core layer thickness on the $D_{core}$ (**a**) and $D_{bottom}$ (**b**) of the mattress at different densities. Significant differences between different letters in the graph are the result of multiple comparisons (*p* < 0.05).

**Figure 15 polymers-15-03681-f015:**
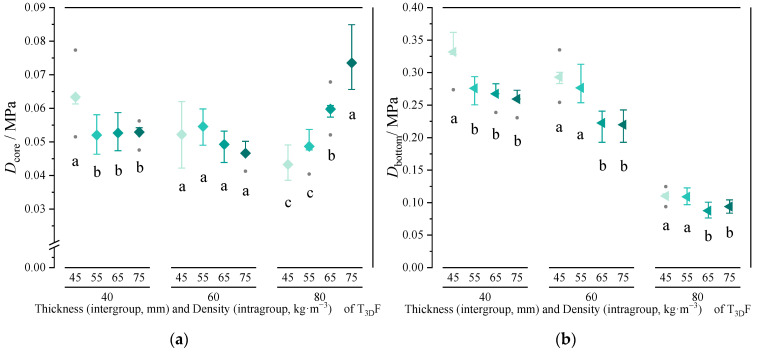
Effect of core layer density on $D_{core}$ (**a**) and $D_{bottom}$ (**b**) of the mattress under different thicknesses. Significant differences between different letters in the graph are the result of multiple comparisons (*p* < 0.05).

**Figure 16 polymers-15-03681-f016:**
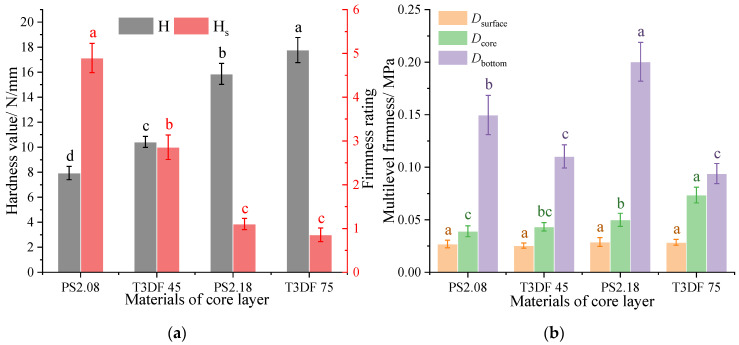
Mattress firmness of different core layers. (**a**) Total firmness, (**b**) Multilevel firmness. Significant differences between different letters in the graph are the result of multiple comparisons (*p* < 0.05).

**Table 1 polymers-15-03681-t001:** Parameters of mattress materials.

Material	Density (kg·m^−3^)	Other Parameters	Thickness (mm)	Mattress Structure
TPEE-3D fibrous material (T_3D_F)	45\55\65\75	-	20	Padding layer
			40	Padding layer/Core layer
			60	Core layer
			80	Core layer
Latex foam (LF)	25	80% latex content	40	Padding layer
Polyurethane foam (PU)	25	-	40	Padding layer
Pocketed spring (PS)	-	2.0 60 6 ^1^	80	Core layer
		2.1 60 6		

^1^ The pocketed spring factors 2.0 60 6 refer to the wire diameter, diameter, and the number of turns of the spring, respectively.

**Table 2 polymers-15-03681-t002:** Indentation properties of T_3D_F.

Density/kg·m^−3^	25% IFD ^1^/N	40% IFD/N	65% IFD/N	*S_f_* ^2^
45	81.8 ± 8.8 ^3^	127 ± 12	359 ± 35	4.4 ± 0.6
55	89.3 ± 8.2	131 ± 11	334 ± 24	3.7 ± 0.3
65	119 ± 14	182 ± 20	361 ± 28	3.0 ± 0.2
75	218 ± 15	334 ± 33	611 ± 82	2.8 ± 0.3

^1^ IFD is the indentation force deflection at n% compression. ^2^ *S_f_* is compression deflection coefficient. ^3^ Data are mean ± standard deviation (SD).

**Table 3 polymers-15-03681-t003:** Indentation properties of the materials.

Material	25% IFD/N	40% IFD/N	65% IFD/N	*S_f_*
PU	132 ± 11	155 ± 13 a ^1^	320 ± 27	2.4 ± 0.2 b
LF	69.3 ± 6.5	134 ± 11 b	386 ± 30	5.6 ± 0.7 a
T_3D_F 45	81.8 ± 8.8	127 ± 12 b	359 ± 35	4.4 ± 0.6 a

^1^ Significant difference between different letters in the same column were the result of multiple comparisons (*p* < 0.05).

**Table 4 polymers-15-03681-t004:** Summary of analysis of variance (ANOVA) results for mattress firmness performed on the density and thickness of T_3D_F padding layers.

Source	*F*-Value	*p*-Value
Density	16.222	<0.001
Thickness	30.583	<0.001
Density × Thickness	1.446	0.148

**Table 5 polymers-15-03681-t005:** Results of main effects of density and thickness of T_3D_F padding layers on each mattress firmness index.

	Density		Thickness	
	*F*-Value	*p*-Value	*F*-Value	*p*-Value
Hardness value	6.592	<0.001	8.061	<0.001
Firmness rating	7.125	<0.001	8.410	<0.001
$D_{surface}$	**9.834** ^1^	<0.001	**14.849**	0.001
$D_{core}$	1.865	0.155	1.274	0.267
$D_{bottom}$	9.354	<0.001	9.044	0.005

^1^ Values in bold indicate the maximum *F*-value in the same column.

**Table 6 polymers-15-03681-t006:** Summary of analysis of variance (ANOVA) results for mattress firmness performed on the density and thickness of the T_3D_F core layer.

Source	*F*-Value	*p*-Value
Density	22.220	<0.001
Thickness	208.981	<0.001
Density × Thickness	7.543	<0.001

**Table 7 polymers-15-03681-t007:** Results of the main effects of density and thickness of the T_3D_F core layer on each mattress firmness index.

	Density		Thickness		Density × Thickness	
	*F*-Value	*p*-Value	*F*-Value	*p*-Value	*F*-Value	*p*-Value
Hardness value	3.117	0.024	165.874	<0.001	1.948	0.319
Firmness rating	8.460	<0.001	120.692	<0.001	2.156	0.076
$D_{surface}$	8.241	0.037	262.161	0.007	0.829	0.554
$D_{core}$	3.065	<0.001	5.568	<0.001	**15.625**	<0.001
$D_{bottom}$	**24.824** ^1^	<0.001	**483.464**	<0.001	3.860	0.003

^1^ Values in bold indicate the maximum *F*-value in the same column.

## Data Availability

All data generated or analyzed during this study are included in this published article (and its Appendix A).

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
