# Peer review of "Compression Property of TPEE-3D Fibrous Material and Its Application in Mattress Structural Layer"

_polymers, 2023, doi:10.3390/polym15183681_

Round 1

Reviewer 1 Report

The paper needs minor revision.

1. In the title "TPEE-3D fiber" is wrong. It should be 3D fabric or 3D fibrous matt.

2. If the densities of the materials (latex, polyurethane) are different, how can their compressional performance be compared? Did you normalize by dividing the values by density?

3. modulus of elasticity is never abbreviated as (MOE). Please check literature.

4. Please use scientific abbreviations and units as in the literature. These are well known parameters and units.

5. Editing may be done with native and professional English speaker.

Editing may be done with native and professional English speaker.

Author Response

Dear reviewer, I have implemented the requested changes. The details are explained in the comment responses and the highlighted section of the paper.

Reviewer 2 Report

Reviewers' comments:

Comments: 
The manuscript reported on
Compression property of TPEE-3D fiber and its application in mattress structural layer. The manuscript needs a detailed editing. The authors need to provide answers to the issues listed below before the manuscript should be accepted for publication.

- In the Abstract: Qualitative information’s are missing in abstract.

- Add more keywords.

- Introduction part should be detailed; it is useful for new readers.

- Please provides the references for equations and formula.

- 2.2.2. Testing of mattress firmness – should be improve.

- 3.1.2. Comparison of the properties between T3DF and other materials - should be improve.

- 3.2.2. Comparison of mattress firmness between T3DF and other padding materials - should be improve.

- Conclusion should be specific short results.

- References: make all references in same format for volume number, page number and journal name, because it is difficult to searching and reading.

- Some sentences need reconstruction and the level of English should be improved.

- Add the graphical abstract, it is use full to readers.

So that I recommended this manuscript to major revision and for future process.

Some sentences need reconstruction and the level of English should be improved.

Author Response

(The authors gave the same response as above.)

Reviewer 3 Report

Dear Authors,

in your interesting manuscript, the following points should be added/changed to further improve it:

- Abstract: Please add the abbreviation TPEE after the full name. How can "3D fibers" be imagined?

- Text below Fig. 2: Here we really need an explanation of the "T3DF". Actually this is not a fiber, but probably a nonwoven, according to Fig. 2b.

- 2.1: Please explain the order of the three parts. Besides, Table 1 is hard to interpret - how many different kinds of mattresses (and how many specimens per sample) were prepared? Why are there four parts of the TPEE mattress while only three parts are mentioned in the text?

- 2.2.1: The sample size is 80 mm x 80 mm x 80 mm, else the unit wouldn't fit. The pressing rate needs a proper multiplication sign and a superscripted "-1". (ditto in lines 121-122, line 133, line 135, line 151 ff)

- Please add references for all equations. In Eq. 4, the K*a in the exponent has a unit, which is not possible. Why is there a factor 10^-6 in Eqs. 6-8? I assume this has to do with the strains, but if all units are correctly taken into account, such factors are normally not correct.

- 2.3: Did you check the prerequisites to allow using an ANOVA?

- 3.1.1: Please define MOE.

- Fig. 5a: Please add the missing spaces between numbers and units. (ditto for the next graphs)

- Fig. 6c: If the inset numbers neab stresses, they need units.

- Tables 2, 3: Standard deviations cannot have more than 2 significant digits, averages have the same accuracy.

- What do the x-error bars in Fig. 7b and the x-variations of the easurement results mean? Or is this just an optical variation meant to enable better visibility of the single dots?

- Fig. 9a: Why do you divide the hardness (firmness) by H (H_s)?

- Fig. 12: The letters/numbers are too small to be properly readable.

- Fig. 13: What do the numbers on the x-axis and on the top mean? Please add units, where necessary. (ditto in Fig. 14)

see above

Author Response

(The authors gave the same response as above.)

Round 2

Reviewer 2 Report

Reviewers' comments:

The authors revised the manuscript according to the reviewers' comments.

So that I recommended this manuscript accept for publication in Polymers.

 Minor editing of English language required

Author Response

Dear Reviewer, we have re-edited the English writing throughout the text.

Reviewer 3 Report

Dear Authors,

in your answer you write "The material of “TPEE-3D fiber” is a fibrous material with a 3D network structure made of TPEE" - please don't tell me, tell the reader. This simple fact is just not available for the reader and thus should be added in the abstract. The additional information from your second answer - that it is not a nonwoven (but sort of sponge?) - would also be helpful for the reader to understand the material. I am especially wondering how the "fibers" are connected, if it is not a nonwoven.

- Eq. 4: Independent from the standard (unfortunately standards sometimes contain errors), let's check the units: K is defined in Eq. 5. A_450 should have the unit N*m or N*mm, H has the unit N/mm, so K has the unit m^2 or mm^2 or any other area. a is dimensionless. It is mathematically simply not possible to have a unit in the exponent.

- Eqs. 6-8: I understand what you want to do with the 10^-6, but this is mathematically wrong. Just leave it away, everybody is able calculate MPa from Pa.

- Table 2: The accuracy of the average must be identical with the accuracy (not the number of significant digits) of the standard deviations, so the first number must be 81.8 +- 8.8, then 89.3 +- 8.2, 119 +- 14, etc.

- Fig. 14: Since the pure numbers are really confusing, I would still suggest to simply add the units to 45, 55 etc.

still okay

Author Response

Dear reviewer, we have revised the paper as requested.
